# An ELISA to Detect Antibodies to Bovine Alphaherpesviruses 1 and 5 and Bubaline Alphaherpesvirus 1 in Cattle Sera

**DOI:** 10.3390/vetsci10020110

**Published:** 2023-02-02

**Authors:** Camila Mengue Scheffer, Sylio Alfredo Petzhold, Ana Paula Muterle Varela, Willian Pinto Paim, Phelipe Magalhães Duarte, Márcia Regina Loiko, Cristine Cerva, Candice Schmidt, Adrieli Wendlant, Samuel Paulo Cibulski, Diane Alves de Lima, Caroline Tochetto, Anne Caroline Ramos dos Santos, Juliana Inês Herpich, Thais Fumaco Teixeira, Helton Fernandes dos Santos, Fabrício Souza Campos, Ana Cláudia Franco, Paulo Michel Roehe

**Affiliations:** 1Programa de Pós-Graduação em Ciências Veterinárias, Faculdade de Medicina Veterinária, Universidade Federal do Rio Grande do Sul, Porto Alegre 90650-002, Brazil; 2Centro de Pesquisa em Saúde Animal, Instituto de Pesquisas Veterinárias Desidério Finamor, Departamento de Diagnóstico e Pesquisa Agropecuária, Secretaria de Agricultura, Pecuária e Desenvolvimento Rural, Eldorado do Sul 90150-004, Brazil; 3Curso de Biomedicina, Campus Primavera do Leste, Universidade de Cuiabá, Cuiabá 78065-900, Brazil; 4Laboratório de Virologia, Departamento de Microbiologia, Imunologia e Parasitologia, Instituto de Ciências Básicas da Saúde, Universidade Federal do Rio Grande do Sul (UFRGS), Porto Alegre 90035-003, Brazil

**Keywords:** serological test, alphaherpesviruses, serology, neutralization, ELISA

## Abstract

**Simple Summary:**

Bovine alphaherpesviruses type 1 (bovine herpesvirus type 1 subtype 1.1, 1.2a, and 1.2b), type 5 (subtypes 5a, 5b, and 5c), and bubaline herpesvirus type 1 (BuHV-1) induce highly, though not fully, cross-reactive antibody responses, but no antibody assay has, to date, been evaluated against all recognized types/subtypes of these viruses. Six hundred bovine field serum samples were screened in serum neutralization tests (SN) performed against the seven virus types/subtypes and tested in seven ELISAs prepared with each of the virus types/subtypes. A combined, multiple antigens ELISA (mAgELISA) was prepared by mixing five viral antigens. In comparison to SN, the mAgELISA sensitivity was 96.5% with 96.1% specificity (κ = 0.93; PPV = 95.0%; NPV = 97.3%). The findings reveal that the mAgELISA developed here is suitable for the detection of antibodies to BoAHV-1, BoAHV-5, and BuHV-1, with sensitivity and specificity comparable to SN, performed with all recognized types of BoAHV-1, BoAHV-5, and BuAHV-1 as challenge viruses.

**Abstract:**

Bovine alphaherpesvirus 1 (subtypes 1.1, 1.2a, and 1.2b), type 5 (subtypes 5a, 5b, and 5c), and bubaline herpesvirus 1 (BuHV-1) induce highly, though not fully cross-reactive serological responses. Most types and subtypes of these viruses circulate particularly in countries of the southern hemisphere, notably Brazil and Argentina. Therefore, the detection of infected animals is important in defining prevention and control strategies, particularly when flocks are destined for international trade. Identification of infected herds is most often achieved by assays that detect antibodies, such as enzyme immunoassays (ELISAs). However, to date, no ELISA has been evaluated in its capacity to detect antibodies to these alphaherpesviruses. Here, an ELISA was developed to detect antibodies to all currently recognized BoAHV-1, BoAHV-5, and BuAHV-1 types/subtypes, and its sensitivity and specificity were determined. Six hundred bovine sera were screened in serum neutralization tests (SN) against the seven viruses. ELISAs prepared with each of the viruses were compared to SN. Subsequently, a combined assay with multiple antigens LISA was prepared by mixing five viral antigens, chosen for their highest sensitivity in the preparative assays. In comparison to SN, the mAgELISA sensitivity was 96.5% with 96.1% specificity (κ = 0.93; PPV = 95.0%; NPV = 97.3%). The findings reveal that the mAgELISA developed here is highly suitable for the detection of antibodies, comparable in sensitivity and specificity to that of SN when performed with all known types and subtypes of bovine and bubaline alphaherpesviruses.

## 1. Introduction

Bovine alphaherpesvirus 1 (infectious bovine rhinotracheitis virus; BoAHV-1), bovine alphaherpesvirus 5 (bovine encephalitis herpesvirus; BoAHV-5), and bubaline alphaherpesvirus 1 (water buffalo herpesvirus; BuAHV-1) are members of the family *Herpesviridae*, subfamily *Alphaherpesvirinae*, genus *Varicellovirus* [1]. Bovine alphaherpesvirus 1 and 5 are currently subdivided into subtypes (BoAHV-1.1, 1.2a, 1.2b; BoAHV-5a, b, c). These are major pathogens of cattle, involved in conditions that may affect the respiratory reproductive and neuronal system [2,3]. BuAHV-1, in its turn, is becoming of greater importance given the rapid expansion of water buffalo or swamp buffalo (*Bubalus bubalis*) farming worldwide. In water buffaloes, BuAHV-1 is believed to induce mostly inapparent infections, although it has been, on occasion, associated with reproductive and respiratory diseases [4,5,6]. A major concern is the cross-species transmission of agents of disease [7,8,9].

At the time of writing of this article, there are 51 complete genomes of BoAHV-1, 6 of BoAHV-5, and 1 BuAHV-1 available at Genbank NCBI. The degree of identity between the three virus types, both at the nucleotide and amino acid levels, is remarkable; BoAHV-1 and BoAHV-5 share a mean overall identity of up to 88% at the nucleotide/amino acid level, whereas BoAHV-5 and BuAHV-1 share a greater than 95% nucleotide/amino acid identity [10]. The antigenic similarity is also intense, which makes the differentiation between these three virus types so cumbersome [11,12].

Ruminant alphaherpesviruses may cross the species barrier [13,14,15,16,17]. The growing number of farms where buffaloes and cattle are raised together or in proximity to each other increases the odds of inter-species transmission of viruses.

No currently available serological assay has been evaluated to detect antibody responses to all of these virus types and subtypes. Here, an immunoassay was developed to detect cross-reactive antibodies to these seven virus types and subtypes, with optimized sensitivity and specificity. A multiple antigen assay (mAgELISA) was prepared with five selected viruses that were shown to display the highest sensitivity when compared to preparative immunoassays and SN performed with the seven distinct challenge viruses.

## 2. Materials and Methods

### 2.1. Collection of Samples

All investigative work involving the collection of samples from animals and subsequent steps in the laboratory was approved by the Ethics Committee (ECAE), Veterinary Research Institute Desidério Finamor (acceptance code: 20/2014).

Six hundred bovine serum samples were collected from herds in ten farms in the state of Rio Grande do Sul, Brazil. The history of vaccination and records on bovine or bubaline herpesvirus infections BoAHV-1 or BoAHV-5 were unknown. Sera were sent to the lab under refrigeration, inactivated at 56 °C for 30 min, and stored at –20 °C until testing.

A pool of sera that reacted positively or negatively at SN in all assays with the seven virus types/subtypes were used as positive/negative controls throughout.

### 2.2. Cells

Madin Darby bovine kidney cells (MDBK; ATCC CCL-22), were multiplied and subcultured every 3–4 days following standard procedures [18]. All cells, sera, and media tested were free of pestiviruses and herpesviruses. These were equally tested free of antibodies to alphaherpesviruses.

### 2.3. Viruses

Representatives of all BoAHV-1, BoAHV-5, and BuAHV-1 types and/or subtypes were used as challenge viruses in serum neutralization assays (SN), and for the preparation of the ELISA antigens. Viral strains used were: BoAHV-1.1 strain Los Angeles, or LA (Genbank access no. MF421714.1) [19], BoAHV-1.2a SV265/96 (partial sequences nos. N173210.1 and KC840313.1) [20,21], BoAHV-1.2b PG1779/03 (Genbank no. DQ173723.1) [2,22], BoAHV-5a EVI88/95 (Genbank nos. KJ143549.1 and KC840325.1) [20], BoAHV-5b A663 (Genbank access no. MW829288.1) [23], BoAHV-5c ISO97/95 [20] and BuAHV-1 b6 (Genbank access no. NC_043054.1) [10]. The viruses were multiplied in MDBK cells and titrated [21].

### 2.4. Serum Neutralization Tests (SN)

The SN tests were performed separately against each of the seven BoAHV-1, BoAHV-5, and BuAHV-1types/subtypes as reported previously [9]. Plates were incubated at 37 °C in a 5% CO_2_ atmosphere for five days and read by examining the presence of cytopathic effect (CPE) in cells. Appropriate negative and positive controls, prepared as described above, were included in each batch of assays.

### 2.5. Antigen Preparation

ELISA antigens were prepared in 850 cm^2^ roller bottles. Each of the viruses was inoculated in nearly confluent monolayers of MDBK cells at a multiplicity of infection equal to 0.01. When the CPE was evident in 80% of the monolayers, the cells were scraped off the flasks, mixed with the supernatant, and centrifuged at 5000× *g* for 15 min. The medium was removed and cells overlaid with 2 mL 0.2% OGP (Sigma#08001) in PBS (0.02% KH_2_PO_4_; 0.09% NaH_2_PO_4_; 0.8% NaCl; 0.02% KCl) and incubated on a rocking platform for 2 h at 4 °C. The suspension obtained was centrifuged at 5000× *g* for 5 min to remove cell debris. This suspension constituted the antigen, which was aliquoted and stored at −70 °C until use. Uninfected cell cultures were similarly prepared and used as control antigens. Viral antigens were prepared with each of the viruses separately.

### 2.6. Optimization of the Assays

Serial dilutions of each of the antigens (1:50 to 1:6400) were prepared in carbonate/bicarbonate buffer (0.3% Na_2_CO_3_, 0.6% NaHCO_3,_ pH 9.6). Each antigen was tested in individual plates (polystyrene microplates, Greiner). The positive and negative serum control pools were diluted (1:2 to 1:4096) in dilution buffer (0.1% Tween 20; 4.0% NaCl; 6.0% bovine serum albumin; 0.8% phenol red) and incubated with each antigen preparation, individually. The peroxidase/anti-bovine IgG conjugate (Sigma-A5295) was titrated and incubated with each antigen preparation. The assays were optimized to provide the most intense differences in optical density (OD) readings between positive and negative sera. The optimum dilutions for the viral antigens to be used in the preparative single (sAg) and multiple antigens (mAg) ELISAs were established at 1:200 (total protein concentration 2.35 ug/uL). Test sera and controls were diluted 1:2. The anti-bovine IgG/peroxidase conjugate was used at 1:1500. Tests were considered valid when the difference in OD readings between positive and negative controls was at least 0.7 OD units. The cut-off point of the ELISAs for each plate was defined at ≥45% of the optical density (OD) of the negative control when the difference between negative and strong positive controls was ≥0.7 OD units. Plates whose readings were not within these limits were repeated.

Sera that gave rise to doubtful results in any of the assays were tested up to three times with the seven preparative “single antigens” ELISAs (sAgELISAs; see below). All sera with OD readings close to the cut-off point were repeated until consistent results were obtained. A negative control antigen prepared with uninfected MDBK cells was added to each battery of assays.

### 2.7. Preparative, “Single Antigens” ELISAs (sAgELISAs)

Preliminary assays consisting of seven preparative, sAgELISAs, each one with a distinct antigen preparation, were carried out; test sera were evaluated separately with each of these. Polystyrene microplates (Greiner) were coated overnight at 4° C with 50 µL of the previously titrated antigens (1:200 in carbonate/bicarbonate buffer). After incubation, plates were washed three times with 100 µL PBST-20 (0.05% Tween 20; 0.02% KH_2_PO_4_; 0.09% NaH_2_PO_4_; 0.8% NaCl; 0.02% KCl, pH 7.3). Subsequently, blocking was performed with 70 µL of 2% milk powder (Molico^®^, Nestlé, Vevey, Switzerland) for 1 h at 37 °C and plates washed as above. Test sera (diluted 1:2) were added to duplicate wells (50 µL/well) and incubated for 1 h at 37 °C. Subsequently, plates were washed again and incubated for 1 h at 37 °C with an appropriate dilution (1:1500 in dilution buffer) of rabbit/anti-bovine peroxidase/IgG conjugate (Sigma-A5295). After another round of washings, 50 µL of the substrate ABTS (Sigma-A1888) with 0.3% H_2_O_2_ was added to the wells. After 30 min of incubation at room temperature, the optical density (OD) was measured (405 nm), and the average OD of each serum was determined. Every batch of tests included positive and negative controls.

### 2.8. Multiple Antigen ELISA (mAgELISA)

The selection of antigens to be included in the mAgELISA was based on the results obtained with the preparative, sAgELISAs. Five of the single antigens that displayed maximum sensitivity at the sAgELISAs (BoAHV-1.1 LA, BoAHV-1.2a SV265/96, BoAHV1.2b PG1779/03, BoAHV-5a EVI 88/95, BoAHV-5c ISO 97/95; Table 1) were combined to prepare the mAgELISA. These were adjusted to a final total protein concentration of 2.35 ug/uL (final dilution 1:200), mixed and added to wells in 50 µL volumes, then incubated overnight at 4 °C. All other procedures were as carried out as described for the sAgELISAs above.

### 2.9. Determination of the Cut-Off Points of sAgELISAs and mAgELISA

For determination of the cut-off points for the ELISAs, 100 sera were used, comprising 80 sera positive in SN tests with the seven alphaherpesviruses examined in this study. Sera with different levels of reactivity at ELISA (high, medium, and weak ELISA-reactive sera), as well as 20 sera negative in all assays, were included. The cut-off point of the ELISAs was established for each plate by comparing the OD of negative controls and strong positive controls.

### 2.10. Sensitivity and Statistical Analyses

The sensitivity of the SN and the seven sAgELISAs was defined by taking the sum of the sera with positive results as the reference. All statistical procedures performed here were calculated using the Dag Stat program [24].

## 3. Results

### Comparisons between SN, sAgELISA, and mAgELISA

From the 259 neutralizing antibody-positive sera, 213 (82.2%) neutralized all of the seven viruses tested. However, the remaining 46 (17.8%) presented discrepant profiles of reactivity. Such variation was dependent on the virus used in the challenge at SN (Figure 1).

The results of SN with all seven viruses, as well as those of the seven sAgELISAs, are shown in Table 1. Out of the 600 sera selected, 259 (43.2%) are positive for neutralizing antibodies to at least one of the challenge viruses, whereas 341 (56.8%) sera are negative for antibodies to all viruses.

In the preparative sAgELISAs (Table 2), 263/600 (43.8%) sera are positive in at least one of the assays with the seven antigens, of which 219/263 (83.3%) are positive with all seven antigens. The remaining 44 (16.7%) give rise to discrepant profiles of reactivity, depending on the antigen used in the sAgELISA. Based on those results, the combination of the five antigens (BoAHV-1.2a, 1.2b, and BoAHV-5a, b, c) that detected most sera as positive in the sAgELISAs was used to prepare the mAgELISA (Table 1).

In the comparison between the results of the seven preparative sAgELISAs and mAgELISA (Table 3), the two assays reveal an almost identical performance, with only two serum samples displaying discrepant results out of the 600 sera examined.

With the mAgELISA, 263 of the total 600 samples give rise to positive results. When compared to the SN (Table 4), the mAgELISA displays 96.5% sensitivity and 96.1% specificity (κ = 0.93; PPV = 95.0%; NPV = 97.3%). Twenty-two sera show discrepant results between the mAgELISA and SN.

## 4. Discussion

Several ELISAs for antibodies to bovine alphaherpesviruses have been developed, most of those based on a single BoAHV-1 strain as the source of antigen [25,26,27,28,29,30]. In Brazil, bovine alphaherpesviruses of all types/subtypes are prevalent [2,20,23,31,32]. In most other countries, such information is not available, that is, the prevalent subtypes of such alphaherpesviruses have not yet been reported. Regarding BuHV-1, although to date it has only been identified in a few countries, is most likely circulating wherever buffaloes are raised. The expansion of bubaline farming worldwide highlights the importance of BuHV-1, particularly given the possibility of cross-species transmission of potentially pathogenic microorganisms [3,7].

In serological diagnosis, the cross-reactivity of BuHV-1-induced antibodies with antibodies to bovine herpesviruses has been documented on several occasions [9,31,33], and is an additional concern wherever bovine and bubaline cattle are raised in proximity. Therefore, an assay that could detect most seropositive animals would be most adequate as a tool to control herpesvirus infections in herds. As such, the objective of the present study was to develop a serological screening test capable of detecting antibodies to bovine and bubaline herpesviruses that may induce cross-reacting antibodies in cattle [12]. To achieve that, an ELISA was developed based on multiple herpesvirus antigens, capable of detecting antibodies to all known types/subtypes of BoAHV-1, BoAHV-5, and BuHV-1. The assay is shown to display sensitivity, specificity, and predictive values comparable to the “gold standard” SN. The validating assays were performed against all targeted herpesviruses types/subtypes. To avoid errors introduced due to the usage of a single virus at the challenge in SN, seven different challenge viruses were used with a 24 h incubation period before the addition of cells, targeting maximum sensitivity [34].

The best combination of antigens for the mAgELISA was achieved by mixing five different antigens. The combined antigens considerably increase the sensitivity of the assay. However, antigen preparation is cumbersome and requires absolute care to avoid cross-contaminations. Nevertheless, the mAgELISA achieves the objective of detecting significantly more seropositive samples than any of the ELISAs prepared with antigens of each of the particular viruses.

Some of the drawbacks raised during the development of the mAgELISA reported here must be commented on. The vast majority of the assays conducted in diagnostic laboratories in search for antibodies to bovine alphaherpesviruses—whether ELISAs or SN—are performed with one strain of BoAHV-1 as a source of antigen or as a challenge virus, often using one of the “classical” BoAHV-1.1 strains. As demonstrated here, depending on the strain used as a source of antigen (or challenge virus at SN), a varying number of false negative results may be obtained. However, maximum sensitivity should be required to minimize the chances of disseminating herpesviral infections, since seropositive animals are potential spreaders of the viruses and may compromise control or eradication efforts. Therefore, an assay capable of detecting most, if not all, seropositive animals becomes an important tool to prevent virus spread. As such, the mAgELISA reported here may become an exquisite tool to minimize the dissemination of BoAHV-1, BoAHV-5, and BuAHV-1.

In addition, particularly in tropical countries, the number of buffalo breeding farms is increasing, in most instances close to, or in conjunction with, bovine cattle farming. The extensive antibody cross-reactivity induced by infections with any of the herpesviruses examined here has already been demonstrated, both here and in previous studies [9,33,35]. Moreover, cross-reactive antibodies may interfere with the trade and exports of live animals, where some countries require the animals to be fully seronegative to BoAHV-1. Thus, cross-reactivity may become problematic. Therefore, it is of significant importance to check the assay’s performance, not only antibodies to BoAHV-1 and BoAHV-5, but also BuAHV-1, since the latter may also induce cross-reactivity in bovine cattle.

However, as currently available serological assays do not discriminate between serological responses induced by these types/subtypes of viruses, determining the virus that induces a particular antibody response is still serologically unattainable.

Another limitation of the assay introduced here is that it does not allow for differentiation between infected and vaccinated animals. This would be essential in countries where differential vaccines (DIVA) are in use. Nonetheless, in Brazil and many other countries, DIVAs are not licensed for BoAHV-1 control; as such, a serological assay that would be adequate for strategies based on the use of DIVAs would not be adequate for such countries. In fact, in the country of origin of the authors and many others, neither DIVAs nor differential antibody assays are licensed for sale. For that reason, we did not perform comparative assays with DIVA-related assays. As such, the assay reported here may not be useful in countries where DIVA-based eradication programs have been implemented. Nevertheless, examining DIVA-compatible assays in their capacity to recognize antibodies induced by infections with all seven herpesvirus types and subtypes, as performed in the present study, remains a future work to be carried out.

In conclusion, it is shown here that the mAgELISA is suitable for the detection of antibodies to BoAHV-1, BoAHV-5, and BuHV-1, with sensitivity and specificity significantly compared to that of at least three SN tests performed with three different challenge viruses. The assay is capable of detecting antibodies to all known types and/or subtypes of BoAHV-1, BoAHV-5, and BuHV-1. To date, the mAgELISA is the first assay to be evaluated in its capacity to detect antibodies to all these alphaherpesviruses. It is expected that this test will become useful in providing support to control or eradicate programs for herpesviruses in ruminants. Moreover, the assay presented here may become useful in the trade of ruminants whenever serological testing of bovine/bubaline herpesviruses may be required.

## Figures and Tables

**Figure 1 vetsci-10-00110-f001:**
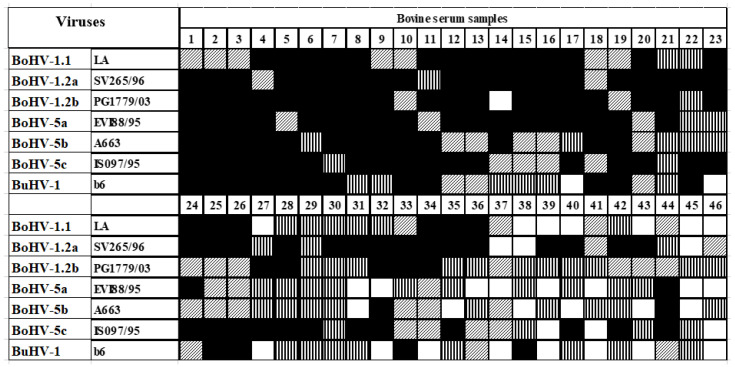
Reactivity profile of neutralizing antibody-positive bovine sera at SN and preparative single antigen ELISAs (sAgELISAs) with seven ruminant alphaherpesviruses as challenge viruses or antigens. Shown only those with discrepant results in any of the assays (n = 46). Black squares: positive reaction to a particular strain in both tests; white squares: negative reaction to a particular strain in both tests; vertical stripes: positive reaction to a particular strain at SN only; diagonal stripes: positive reaction to a particular strain at the sAgELISA only.

**Table 1 vetsci-10-00110-t001:** Serum neutralization tests (SN) with bovine and bubaline alphaherpesviruses and the single antigen ELISAs (sAgELISA) (n = 259).

Virus Strains	SN		sAgELISAs	
	Number of Positive Sera	Sensitivity ^a^	Number of Positive Sera	Sensitivity ^a^
BoAHV-1.1 (LA)	236	91.1	237	91.5
BoAHV-1.2a (SV265/96)	244	94.2	247	95.4
BoAHV-1.2b (PG1779/03)	241	93.1	241	93.1
BoAHV-5a (EVI88/95)	239	92.3	233	90.0
BoAHV-5b (A663)	235	90.7	232	89.6
BoAHV-5c (ISO97/95)	242	93.4	249	96.1
BuAHV-1 (B6)	237	91.5	229	88.4
LA + SV265/96	248	95.8	251	96.9
LA + PG1779/03	249	96.1	246	95.0
LA + EVI88/95	251	96.9	240	92.7
LA + A663	249	96.1	239	92.3
LA + ISO97/95	250	96.5	251	96.9
LA + B6	249	96.1	240	92.7
SV265/96 + PG1779/03	250	96.5	254	98.1
SV265/96 + EVI88/95	249	96.1	251	96.9
SV265/96 + A663	249	96.1	251	96.9
SV265/96 + ISO97/95	249	96.1	257	99.2
SV265/96 + B6	250	96.5	252	97.3 ^&^
PG1779/03 + EVI88/95	250	96.5	243	93.8
PG1779/03 + A663	247	95.4	243	93.8
PG1779/03 + ISO97/95	251	96.9	255	98.5 ^&^
PG1779/03 + B6	249	96.1	246	95.0
EVI88/95 + A663	246	95.0	237	91.5
EVI88/95 + ISO97/95	249	96.1	252	97.3 ^&^
EVI88/95 + B6	247	95.4	239	92.3
A663 + ISO97/95	249	96.1	251	96.9
A663 + B6	247	95.4	238	91.9
ISO97/95 + B6	249	96.1	252	97.3 ^&^
LA + SV265/96 + PG1779/03	253	97.7 ^&^	255	98.5 ^&^
LA + SV265/96 + EVI88/95	252	97.3 ^&^	253	97.7 ^&^
LA + SV265/96 + A663	252	97.3 ^&^	252	97.3 ^&^
LA + SV265/96 + ISO97/95	252	97.3 ^&^	257	99.2 ^&^
LA + SV265/96 + B6	253	97.7 ^&^	253	97.7 ^&^
LA + PG1779/03 + EVI88/95	255	98.5 ^&^	247	95.4
LA + PG1779/03 + A663	253	97.7 ^&^	247	95.4
LA + PG1779/03 + ISO97/95	255	98.5 ^&^	256	98.8 ^&^
LA + PG1779/03 + B6	253	97.7 ^&^	249	96.1
LA + EVI88/95 + A663	254	98.1 ^&^	242	93.4
LA + EVI88/95 + ISO97/95	254	98.1 ^&^	253	97.7 ^&^
LA + EVI88/95 + B6	254	98.1 ^&^	243	93.8
LA + A663 + ISO97/95	255	98.5 ^&^	252	97.3 ^&^
LA + A663 + B6	255	98.5 ^&^	242	93.4
LA + ISO97/95 + B6	254	98.1 ^&^	253	97.7 ^&^
LA + SV265/96 + PG1779/03 + EVI88/95	255	98.5 ^&^	256	98.8
LA + SV265/96 + PG1779/03 + A663	254	98.1 ^&^	256	98.8
LA + SV265/96 + PG1779/03 + ISO97/95	255	98.5 ^&^	259	100 ^&^
LA + SV265/96 + PG1779/03 + B6	255	98.5 ^&^	257	99.2
LA + PG1779/03 + EVI88/95 + A663	255	98.5 ^&^	248	95.8
LA + PG1779/03 + EVI88/95 + ISO97/95	257	99.2 ^&^	256	98.8 ^&^
LA + PG1779/03 + EVI88/95 + B6	257	99.2 ^&^	250	96.5
LA + EVI88/95 + A663 + ISO97/95	257	99.2 ^&^	254	98.1 ^&^
LA + EVI88/95 + A663 + B6	257	99.2 ^&^	245	94.6
LA + A663 + ISO97/95 + B6	258	99.6 ^&^	254	98.1 ^&^
SV265/96 + PG1779/03 + EVI88/95 + A663	252	97.3 ^&^	256	98.8 ^&^
SV265/96 + PG1779/03 + EVI88/95 + ISO97/95	254	98.1 ^&^	259	100 ^&^
SV265/96 + PG1779/03 + EVI88/95 + B6	254	98.1 ^&^	258	99.6 ^&^
SV265/96 + EVI88/95 + A663 + ISO97/95	254	98.1 ^&^	259	100 ^&^
SV265/96 + EVI88/95 + A663 + B6	254	98.1 ^&^	256	98.8 ^&^
SV265/96 + A663 + ISO97/95 + B6	255	98.5 ^&^	259	100 ^&^
PG1779/03 + EVI88/95 + A663 + ISO97/95	254	98.1 ^&^	256	98.8 ^&^
PG1779/03 + EVI88/95 + A663 + B6	254	98.1 ^&^	249	96.1
PG1779/03 + A663 + ISO97/95 + B6	255	98.5 ^&^	258	99.6 ^&^
EVI88/95 + A663 + ISO97/95 + B6	255	98.5 ^&^	255	98.5 ^&^
LA + SV265/96 + PG1779/03 + EVI88/95 + A663	255	98.5 ^&^	257	99.2 ^&^
**LA + SV265/96 + PG1779/03 + EVI88/95 + ISO97/95 ****	**257**	**99.2 ^&^**	**259**	**100 ^&^**
LA + SV265/96 + PG1779/03 + EVI88/95 + B6	257	99.2 ^&^	258	99.6 ^&^
LA + PG1779/03 + EVI88/95 + A663 + ISO97/95	257	99.2 ^&^	257	99.2 ^&^
LA + PG1779/03 + EVI88/95 + A663 + B6	257	99.2 ^&^	251	96.9
LA + EVI88/95 + A663 + ISO97/95 + B6	259	100 ^&^	256	98.8 ^&^
SV265/96 + PG1779/03 + EVI88/95 + A663 + ISO97/95	254	98.1 ^&^	259	100 ^&^
SV265/96 + PG1779/03 + EVI88/95 + A663 + B6	254	98.1 ^&^	259	100 ^&^
PG1779/03 + EVI88/95 + A663 + ISO97/95 + B6	256	98.8 ^&^	258	99.6 ^&^
LA + SV265/96 + PG1779/03 + EVI88/95 + A663 + SO97/95	257	99.2 ^&^	259	100 ^&^
LA + SV265/96 + PG1779/03 + EVI88/95 + A663 + B6	257	99.2 ^&^	259	100 ^&^
SV265/96 + PG1779/03 + EVI88/95 + A663 + ISO97/95 + B6	256	98.8 ^&^	259	100 ^&^
LA + SV265/96 + PG1779/03 + EVI88/95 + A663 + ISO97/95 + B6	259	100.0	259	100 ^&^

^a^ Sensitivity calculated over the highest number of SN-positive sera (SN = 259). ^&^ Sensitivity above 97.3 (252/259 samples) is significant by McNemar’s test. ** The combination of antigens used for preparing the mAgELISA is bolded.

**Table 2 vetsci-10-00110-t002:** Comparison between serum neutralization (SN) and sAgELISAs for detection of antibodies to alphaherpesviruses in bovine sera (n = 600).

SN	sAgELISA	TOTAL
POSITIVE	NEGATIVE
POSITIVE	250(a)	9(b)	259(a + b)
NEGATIVE	13(c)	328(d)	341(c + d)
TOTAL	263(a + c)	337(b + d)	600(a + b + c + d)
Sensitivity (a/a + c) × 100 = 96.56%; Specificity (d/d + b) × 100 = 96.1%;Positive predictive value (PPV) (a/a + b) × 100 = 95%; Negative predictive value (NPV) (d/c + d) × 100 = 97.3%; Kappa correlation:
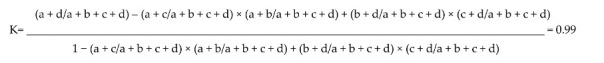 McNemar’s = [(a − d)/−1]2/a + d = significant at *p* > 0.05.

**Table 3 vetsci-10-00110-t003:** Comparison between results of the sAgELISAs and the mAgELISA for detection of antibodies to bovine and bubaline alphaherpesviruses in bovine sera (n = 600) **.

sAgELISA *	mAgELISA		TOTAL
	POSITIVE	NEGATIVE	
POSITIVE	262	1	263
NEGATIVE	1	336	337
TOTAL	263	337	600

* Positive in at least one of the seven sAgELISAs; ** Calculations as in Table 2; Sensitivity = 99.6%; Specificity = 99.7%; PPV = 99.6%; NPV = 99.7%; K = 0.99. McNemar’s: significant at *p* > 0.05.

**Table 4 vetsci-10-00110-t004:** Comparison between serum neutralization (SN) and mAgELISA in detecting antibodies to bovine and bubaline alphaherpesviruses in bovine sera (n = 600) **.

SN	mAgELISA		TOTAL
	POSITIVE	NEGATIVE	
POSITIVE	250	9	259
NEGATIVE	13	328	341
TOTAL	263	337	600

** Calculations as in Table 2. Sensitivity = 96.5%; Specificity = 96.1%; PPV = 95%; NPV = 97.3%; K = 0.93; McNemar’s: significant at *p* > 0.05.

## Data Availability

Data is contained within the article.

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
