# Peer review of "An ELISA to Detect Antibodies to Bovine Alphaherpesviruses 1 and 5 and Bubaline Alphaherpesvirus 1 in Cattle Sera"

_vetsci, 2023, doi:10.3390/vetsci10020110_

Round 1

Reviewer 1 Report

The manuscript entitled " An enzyme-linked immunosorbent assay (ELISAs) to detect antibodies to bovine alphaherpesviruses types 1 (BoHV-1) and 5 (BoHV-5) and bubaline alphaherpesvirus 1 (BuHV-1) in cattle sera” in which, the authors have developed an ELISA-based serological assay for the detection of Bovine alphaherpesviruses type 1 (bovine herpesvirus type 1 subtypes 1.1, 1.2a, and 1.2b), type 5 (subtypes 5a, 5b, and 5c), and bubaline herpesvirus type 1 (BuHV-1), and its sensitivity and specificity were determined by comparing with serum neutralization tests (SN). The study is interesting. However, there are several issues that need to be addressed before considering the manuscript for publication.

Major corrections:

1.       The current ELISA described cannot differentiate the infected animals (BoHV-1, BoHV-5, and BuHV-1)  from the vaccinated. This is the major drawback of the study. Why did the authors, not attempt to develop the ELISA which supports the DIVA strategy, or keep it in the account?

2.       Introduction section needs to be improved significantly. Authors should discuss the antigenic similarity and sequence homology between types and subtypes (BoHV-1, BoHV-5, and BuHV-1).

3.       The authors should give the concentrations of each viral antigen for ELISA in terms of µg/ng/pg per ml or per well instead of mentioning 1:200 dilutions. It will help to repeat the assays by other investigations.

4.       Did the authors use known positive and known negative serum samples for each of BoHV-1, BoHV-5, and BuHV-1? If not, it is necessary to include them in the ELISA standardization procedure.

5.       Did the authors validate the assay by performing the ELISA in at least 2 or 3 different labs, physically? Which is necessary for validating the diagnostic assay.

6.       What is the prevalence of BuHV-1 among the cattle population? What is the significance of including the BuHV-1 in the antigenic mixture for ELISA? Differentiating the type of infection (between BoHV-1, BoHV-5, and BuHV-1) would be more appropriate and logical. The authors should discuss this in detail.

7.       Why did the authors, not include sera from buffalo to test the cross-reactivity between different viral antigens in ELISA?

8.       The authors have not provided the Genbank accession numbers for all viral strains used. It should be included in the manuscript.

9.       A paragraph describing the study's potential limitations: discuss in detail the limitation of the current ELISA in terms of non-suitability for the DIVA strategy (differentiating infected from vaccinated animals).

Minor corrections:

·         Manuscript should be revised for the English language.

·         Figure 1: adding an additional column mentioning the type or subtype of the virus would help better understanding.

·         Table 1: the current version of the table is very confusing. The authors should consider reformatting the table to make it easy to follow.

Author Response

Reviewer 1

Major corrections:

  1. The current ELISA described cannot differentiate the infected animals (BoHV-1, BoHV-5, and BuHV-1) from the vaccinated. This is the major drawback of the study. Why did the authors, not attempt to develop the ELISA which supports the DIVA strategy, or keep it in the account?

Dear reviewer, thank you for your most appropriate question. The aim of the work was not to develop an assay that would differentiate vaccinated from infected animals. The target was  to develop an assay that could detect the majority of seropositive cattle to any of the recognized types of BoHV-1, BoHV-5, and BuHV-1. In Brazil and many other countries, DIVAs are not licensed for BoHV-1 control; as such, a serological assay that would be used to differentiate infected from vaccinated animals would be of no use in such countries. In Brazil, neither differential vaccines nor differential antibody assays are licensed for sale, which is the main reason we did not perform comparative tests with this type of assay.  Once this paper is published, most likely we will find partners to perform comparative assays with the mAg ELISA and DIVA-antibody assays in countries where the DIVA strategy is used.

  1. Introduction section needs to be improved significantly. Authors should discuss the antigenic similarity and sequence homology between types and subtypes (BoHV-1, BoHV-5, and BuHV-1).

Thank you for your suggestion.  We added a paragraph on the degree  of nucleotide and antigenic similarity between the viruses in lines 65-73.

  1. The authors should give the concentrations of each viral antigen for ELISA in terms of µg/ng/pg per ml or per well instead of mentioning 1:200 dilutions. It will help to repeat the assays by other investigations.

Thank you for your observation. The protein concentration of antigen in crude viral antigen preparations for ELISAs might be somewhat misleading, in that the protein concentration actually will tell you the total concentration of protein, not viral proteins. The amount of viral protein is probably a tiny fraction of the total protein concentration. Thus, defining a protein concentration might be misleading, rather than helpful to someone wishing to repeat the assay. Titrating the antigens is essential for a successful assay. Nevertheless, we have determined the total protein concentration of the antigens, which was 2,35 ug/uL. This information was included in the article.

  1. Did the authors use known positive and known negative serum samples for each of BoHV-1, BoHV-5, and BuHV-1? If not, it is necessary to include them in the ELISA standardization procedure.

Thank you for your question. Yes, known samples with previously defined antibody titers – positive and negative – to every one of the seven viruses, were determined. Those which were positive in all assays were pooled to make positive controls; those which were fully negative in all assays were pooled to make negative controls. The modification is in the article, now on lines 130-132.

  1. Did the authors validate the assay by performing the ELISA in at least 2 or 3 different labs, physically? Which is necessary for validating the diagnostic assay.

No, unfortunately, we did not perform the ELISA in two or three different labs. We, however, performed the ELISA with at least three distinct operators within our lab. Many of the assays were repeated even more than three times.   

  1. What is the prevalence of BuHV-1 among the cattle population? What is the significance of including the BuHV-1 in the antigenic mixture for ELISA? Differentiating the type of infection (between BoHV-1, BoHV-5, and BuHV-1) would be more appropriate and logical. The authors should discuss this in detail.

Thank you for your question. The prevalence of BuHV-1 in the bovine cattle population is unknown since the vast majority of the assays conducted in diagnostic labs to determine antibodies to bovine alphaherpesviruses are performed with one strain of BoHV-1 as the challenge virus, often using one of the “classical” BoHV-1.1 strains. So, we are sorry, but this question remains unanswered. We expect that, as typing and subtyping become more widely practiced, this question might be solved.

BuHV-1 was included in the assays because the number of buffalo breeding farms is increasing in tropical/subtropical countries, in most instances close to, or in conjunction with, bovine cattle farming. Therefore, it became important to check the assay’s performance at the detection of anti-BuHV-1 antibodies, since these may also induce serological crossreactivity in bovine cattle, and vice-versa. Cross-reactive antibodies may interfere not only with prevention and control attempts but also with exports of live animals, where some countries require the animals to be fully seronegative to BoHV-1.

Regarding the differentiation between types of infection, currently available serological assays do not discriminate serological responses induced by these types/subtypes. So, unfortunately, the definition of which virus induced a particular antibody response is still serologically unattainable.  This can only be done by extracting viral DNA from the neuronal ganglia of animals at slaughter, such as we have done in:

Campos et al. 2009 (High prevalence of co-infections with bovine herpesvirus 1 and 5 found in cattle in southern Brazil. Veterinary Microbiology (Amsterdam), v. 138, p. 10.1016/j.vetmi),

and in:

Puentes, R. et al. 2016. Comparison between DNA Detection in Trigeminal Nerve Ganglia and Serology to Detect Cattle Infected with Bovine Herpesviruses Types 1 and 5. Plos One, v. 11, p. e0155941.

  1. Why did the authors, not include sera from buffalo to test the cross-reactivity between different viral antigens in ELISA?

Thank you for your question. We did test the mAgELISA with bubaline sera. This is the subject of another paper, which was just recently submitted to another journal. We can say that serological crossreactivity in bubaline cattle is very similar to that displayed by bovine cattle.

  1. The authors have not provided the Genbank accession numbers for all viral strains used. It should be included in the manuscript.

Thank you for your observation. We have included the access codes for those strains which have already been sequenced, either fully, or partially. These were included in the article in lines 112 -117.

  1. A paragraph describing the study's potential limitations: discuss in detail the limitation of the current ELISA in terms of non-suitability for the DIVA strategy (differentiating infected from vaccinated animals).

Thank you for your suggestion. We have included a paragraph about the study’s potential limitations on lines 292-328 in the revised version of the article.

Minor corrections:

Manuscript should be revised for the English language.

Thank you for your advice. The manuscript was revised for the English language.

Figure 1: adding an additional column mentioning the type or subtype of the virus would help better understanding.

Thank you for the observation. Figure 1 was improved to make it clearer to the readers.

Table 1: the current version of the table is very confusing. The authors should consider reformatting the table to make it easy to follow.

Thank you for the suggestion. We have made some changes to Table 1 to make it more reader-friendly.

Reviewer 2 Report

In this study, the authors developed a mAgELISA to detect antibodies to BoHV-1, BoHV-2 and BuHV-1. I have a few concerns regarding the results.

1. Line 116, please specify what are the negative and positive controls.

2. Line 176, why five not all seven antigens were used to develop mAgELISA?

3. In Table1, why some numbers are in bold? Which values were used for McNemar’s analysis? And p<0.05 means significant for what? Why some numbers are over 259 if only 259 samples were used?

4. In Tables 2/3/4, please specify how sensitivity, specificity, positive predictive value, and negative predictive value were calculated. And the meaning of Kappa and McNemar’s value.

Author Response

Reviewer 2

Comments and Suggestions for Authors

In this study, the authors developed a mAgELISA to detect antibodies to BoHV-1, BoHV-2, and BuHV-1. I have a few concerns regarding the results.

  1. Line 116, please specify what are the negative and positive controls.

Thank you for your question. Known samples with previously defined antibody titers – positive and negative – to every one of the seven viruses, were determined. Those which were positive in all assays were pooled to make positive controls; those which were fully negative in all assays were pooled to make negative controls. The modification on the article is now on lines 130-132.

  1. Line 176, why five, not all seven antigens were used to develop mAgELISA?

Thank you for your question. This was done so because the maximum sensitivity was achieved already with these five viruses (159/159 sera in the mAgELISA with those antigens). We highlighted the combination of antigens used for the mAgELISA in Table 1.

  1. In Table 1, why some numbers are in bold? Which values were used for McNemar’s analysis? And p<0.05 means significant for what? Why some numbers are over 259 if only 259 samples were used?

See below;

  1. In Tables 2/3/4, please specify how sensitivity, specificity, positive predictive value, and negative predictive value were calculated. And the meaning of Kappa and McNemar’s value.

Thank you for your questions 3 and 4. We corrected and modified Table 1 to make it more reader-friendly.

The Kappa index refers to the degree of correlation between two variables, where “1” would be a perfect correlation, and “0”, no correlation. The McNemar index is used to look for differences in two variables that may or may not be significant. It is usually applied on 2 x 2 contingency tables, as is the case in the comparisons performed in this study. In this study, sensitivities above 97,3 (252/259 sera) were significant by McNemar’s test, i.é. detected a significant number of positive sera. This was included in the footnotes of Table 1.

The PPV and NPV are also calculated on 2x2 tables and defined here as the possibility of two assays to give the same (positive or negative) result.

We added a simpler description of the formulae in the article, but we still find these may not be necessary, as the reference for the statistical methods, using the Dag Stat program, is provided (Mackinnon, A. 2000. A spreadsheet for the calculation of comprehensive statistics for the assessment of diagnostic tests and inter-rater agreement. Comput. Biol. Med.30: 127–134. PMID: 10758228).

The authors’ final comment:

We expect to have addressed all your questions. Thank you for your comments and suggestions, which certainly contribute to the improvement of the quality of the paper. If you have any more queries please return to us.

Round 2

Reviewer 1 Report

The authors have addressed all my comments.